# Predicting Health-Related Quality of Life Using Social Determinants of Health: A Machine Learning Approach with the All of Us Cohort

**DOI:** 10.3390/bioengineering12020166

**Published:** 2025-02-09

**Authors:** Tadesse M. Abegaz, Muktar Ahmed, Askal Ayalew Ali, Akshaya Srikanth Bhagavathula

**Affiliations:** 1Division of Pharmacy Practice and Science, College of Pharmacy, The Ohio State University, 281 W Lane Ave, Columbus, OH 43210, USA; 2Adelaide Medical School, Faculty of Health and Medical Sciences, The University of Adelaide, Adelaide, SA 5005, Australia; muktar.ahmed@mymail.unisa.edu.au; 3Economic, Social and Administrative Pharmacy (ESAP), Institute of Public Heath, College of Pharmacy and Pharmaceutical Sciences, Florida A&M University, Tallahassee, FL 32307, USA; askal.ali@famu.edu; 4Department of Public Health, College of Health and Human Services, North Dakota State University, Fargo, ND 58108, USA; akshaya.bhagavathula@ndsu.edu

**Keywords:** machine learning, quality of life, All of Us, social determinants

## Abstract

This study applied machine learning (ML) algorithms to predict health-related quality of life (HRQOL) using comprehensive social determinants of health (SDOH) features. Data from the All of Us dataset, comprising participants with complete HRQOL and SDOH records, were analyzed. The primary outcome was HRQOL, which encompassed physical and mental health components, while SDOH features included social, educational, economic, environmental, and healthcare access factors. Three ML algorithms, namely logistic regression, XGBoost, and Random Forest, were tested. The models achieved accuracy ranges of 0.73–0.77 for HRQOL, 0.70–0.71 for physical health, and 0.72–0.77 for mental health, with corresponding area under the curve ranges of 0.81–0.84, 0.74–0.76, and 0.83–0.85, respectively. Emotional stability, activity management, spiritual beliefs, and comorbidity were identified as key predictors. These findings underscore the critical role of SDOH in predicting HRQOL and suggests future research to focus on applying such models to diverse patient populations and specific clinical conditions.

## 1. Introduction

Health-related quality of life (HRQOL) refers to individuals’ perception of their physical, spiritual, emotional, and mental health. It is an important parameter for evaluating overall health status and the influence of several factors on health outcomes. HRQOL status might vary due to several social and environmental factors [1,2]. Among these factors, social determinants of health (SDOH) play a significant role. SDOH are the conditions and contexts in which people are born, live, learn, play, work, and worship across the lifespan that influence quality of life outcomes [3,4]. Key SDOH include access to safe housing, education, healthcare, nutritious foods, and income as well as exposure to racism, discrimination, and environmental hazards. Research indicates that nearly 50% of the variation in health outcomes can be attributed to SDOH, which is two times higher than the variation caused by the difference in clinical care [5].

Previous studies have used SDOH to predict various health-related outcomes. For example, a study predicted health-related social needs among Medicare and Medicaid beneficiaries using SDOH, achieving moderate prediction accuracy [6]. A recent study utilized SDOH in the All of Us (AoU) dataset to predict depression, delayed medical care, and emergency room visits [7]. Other SDOH features such as levels of optimism on things [8,9], being on top of things, level of education, and occupational status were determinants of HRQOL [10]. Furthermore, it was reported that emotional control was an important predictor of HRQOL, particularly, mental health status [11]. Additionally, studies have demonstrated that four SDOH indicators, such as being in a relationship, level of education, occupational status, and net income per household, were independent predictors of HRQOL [10]. Another study from a representative sample of National Health and Nutrition Examination Survey reported that activities of daily living including bathing or showering, dressing, getting in and out of bed or a chair, walking, using the toilet, and eating were associated with HRQOL [12].

Machine learning (ML) algorithms have emerged as the state-of the-art tools for predicting health outcomes, including HRQOL, owning to their best predictive performance [13]. For instance, the Korean Medicine Daejeon Citizen cohort study utilized ML algorithms using patients’ lifestyle and demographic characteristics to predict HRQOL, achieving an area under curve of 0.82 [14]. Similarly, Liao WW et al., (2022) utilized an ML model in chronic stroke patients [15]. Other studies have applied ML to specific patient populations such as those with Parkinson’s disease and brain tumors, showing varying degree of success in predicting HRQOL [16,17,18,19,20,21]. While these studies are promising, they often involve small sample sizes or limited clinical features, highlighting the need for more comprehensive models. In addition, the implementation of ML models to predict the HRQOL has not been thoroughly studied using key social determinants of HRQOL in a diverse population who were underserved/underrepresented in medical research, which could help to extrapolate the findings in these special populations.

Therefore, the present study has incorporated comprehensive SDOH data from the AoU Research Program to predict HRQOL using ML algorithms. The AoU research program is a large, national dataset of individuals historically underrepresented in medical research. It provides data from electronic health records, physical measurements, Fitbit data, survey, and genomics [22,23,24]. The AoU dataset reports the SDOH and HRQOL of participants. By incorporating diverse SDOH factors and validated ML models with large and diverse populations, the study aimed to develop fair and unbiased algorithms applicable to underserved populations. Furthermore, the incorporation of a comprehensive list of (~80) SDH factors to test their impact on HRQOL would allow a new arena of measuring HRQOL.

This approach not only aligns with the Health People 2030 goals of promoting health equity, but also offers a new paradigm for clinicians to evaluate HRQOL beyond traditional clinical measures [25].

## 2. Materials and Methods

### 2.1. Study Design

A supervised ML approach was employed to predict HRQOL based on SDOH features. Data on HRQOL and SDOH were collected between 1 November 2021 and 30 June 2022 and analyzed from June to November 2024.

### 2.2. Data Source

Data were obtained from the AoU Research Program. Details about the AoU dataset can be found( https://allofus.nih.gov/ Accessed 1 July 2024) [26]. Briefly, the AoU dataset is a large longitudinal dataset comprising electronic health records, surveys, Fitbit data, genetic data, and SDOH. To date, a total of 413,457 participants have enrolled in the AoU Program. Of these, 117,783 responded to the SDH survey, which accounts for 29.6% of the total AoU participants. The SDOH survey, introduced in 2021, was approved by the AoU Institutional Review Board [27] and underwent cognitive interviewing and pilot testing in both English and Spanish to ensure quality.

### 2.3. Population

All participants aged between 18 and 85 with complete records of both SDOH and HRQOL were included. Of the 117,783 individuals who completed the SDOH and overall health survey, 97,175 participants fulfilled the inclusion criteria and were enrolled for model validation. The cohort was predominantly non-Hispanic Whites (>80%), with 91% improved HRQOL, 81.2% improved physical health, and 87% improved mental health. Nearly 10% of the participants had diabetes, 5% had chronic kidney disease, 3.5% had heart failure, and 10% had asthma. Detailed characteristics of the study participants can be found in Appendix A.

### 2.4. Data Processing

Data wrangling and analysis were performed using the R-software. Demographic characteristics, healthcare utilization, SDOH, and medical conditions were merged for analysis. Health status, including HRQOL, physical, and mental health, was initially rated using Likert scale and later transformed into dichotomous variables (improved and unimproved). Different feature engineering techniques were employed to prepare the data for analysis, including outlier detection, the standardization of continuous variables, and one-hot matrix/encoding of categorical variables. Outliers were detected using descriptive statistics and removed based on domain knowledge and data visualization. To address class imbalance, the Random Over-Sampling Examples technique was employed, a bootstrap-based technique which handles categorical data by generating synthetic examples from a conditional density estimate of the two classes [28].

### 2.5. SDOH Features

The SDOH features were our input/predictor variables. The SDOH components in the AoU dataset were developed by team of experts and have demonstrated strong psychometric properties, with a Cronbach’s alpha of 0.80. All SDOH features reported in the AoU dataset were summarized into four domains: social and community context, economic stability, neighborhood and built environment, and health and healthcare. The social and community context constitutes social cohesion among neighbors (4 items), social support (8 items), loneliness (8 items), perceived discrimination (10 items), perceived stress (10 items), daily spiritual experiences (6 items), religious service attendance (1 items), and English proficiency (1 item). The economic stability context comprised food insecurity, housing instability indicator, and housing quality. The neighborhood and built environment domain contains neighborhood physical disorder (6 items), neighborhood social disorder (7 items), neighborhood walkability (5 items), neighborhood crime (2 items), neighborhood residential density (1 item), while the health and healthcare encompass perceived discrimination in medical care settings (7 items). Additionally, features related to healthy affordability, such as inability to afford prescription medications and delays in medication due to cost, were included. All SDOH input variables were discrete with multiclass responses/level. Additional information on the description of the SDOH can also be found at https://www.nature.com/articles/s41598-024-57410-6 (Accessed 1 July 2024) [27].

### 2.6. Health-Related Quality of Life

The self-reported health-related quality of life was used as the outcome variable. The overall health status survey in the AoU dataset contains questions about how participants rate their health. The participants rated their general health, HRQOL physical health, and mental health in a Likert-scale including excellent, very good, good, fair, and poor. The rating was then converted to dichotomous variables by recategorizing the responses into “improved” health status versus “unimproved”. Those patients with excellent, very good, and good responses were categorized as having improved health status, while those with poor and fair were classified as having unimproved health status. The improved category of the HRQOL was used as positive class.

### 2.7. Algorithm Selection and Performance Measures

A supervised ML algorithm was applied using extreme gradient boosting (XGBoost), random forest (RF), and logistic regression (LR). Both XGBoost and RF are tree-based models known for high predictive accuracy and ability to handle large datasets. Tree-based models were selected for their ability to inherently model non-linear relationships and capture complex interactions between variables. These models provide a robust performance baseline due to their flexibility and capability to handle both numerical and categorical data effectively. Additionally, ensemble tree-based methods have demonstrated strong performance in clinical predictive modeling tasks [29].

The dataset was randomly split into two parts: 80% for training and 20% for testing using a stratified splitting/sampling to ensure that the proportion of each class in the training and test sets is the same as that in the original dataset. The ML models were trained to classify the patients into improved and unimproved HRQOL status using the SDOH features. The levels were as follows: 0 indicates no improvement in HRQOL and 1 indicates improved HRQOL. Cross-validation was conducted to evaluate model performance on new data. Specifically, a 5-fold cross-validation was used, where the training data were portioned into five subsets. The model was trained on four subsets while its accuracy was assessed on the remaining portion. This step ensured the model had neither overfit nor underfit. Hyperparameter tuning was performed to improve the accuracy of the ML models. For the RF model, the number of trees (ntree) and the number of variables randomly selected at each split (mtry) were used to tune the model. The ntree refers to the number of trees that are grown within the random forest model. It defines how many individual decision trees are used to make a final prediction. We started with the default value of 500 trees and found the appropriate number that gives us low out-of-bag (OOB) error. On the other hand, mtry refers to the number of variables that are randomly sampled at each split when building a decision tree within the forest. It controls how many variables are considered as candidates for splitting at each node. The default value of mtry is the square root of total number of features and adjusted based on the performance of the model. The improvement in the RF model performance was evaluated using the decrease in out-of-bag (OOB) error, while learning curves were used to evaluate the XGBoost model. Variable importance was assessed through the mean decrease in the accuracy score from the RF model, which indicates how much the model accuracy is affected when a specific variable is excluded [30]. A higher decrease in accuracy in the absence of the variable signals that the variable is more important for the model’s classification accuracy. Model performance was assessed using precision, recall, classification accuracy, F_1_ score, and area under the receiver operating characteristic curve (AUC-ROC). The F_1_ score is a harmonic mean of precision and recall. It effectively balances precision and recall, which makes it a suitable metric for evaluating classification models, particularly in cases where data are imbalanced [31]. The F_1_ score was calculated as follows:(1)$$F_{1}=\frac{2\;*\;Precison\;*\;Recall}{\begin{array}{c} Precison\;+\;Recall \end{array}}\;\;\;Precision=\frac{TP}{TP+FP}\;\;\;\;Recall=\frac{TP}{TP+FN}$$
where TP: true positives; FP: false positive; and FN: false negative.

A higher F_1_ values indicates better model performance. On the other hand, precision refers to the positive predictive value, while recall measures the model’s sensitivity in correctly identifying positive cases. The findings of the present study were reported in compliance with the All of Us Data and Statistics Dissemination Policy disallowing the disclosure of group counts under 20.

## 3. Results

### 3.1. Patient Characteristics

A total of 97,175 participants were included for testing the models, of which 81% were non-Hispanic Whites. About 91% had improved HRQOL, 81.2% had improved physical, and 87% had improved mental health. About 9.4% had DM, 5% CKD, 3.5% HF, and 10.5% a history of asthma (Appendix A).

### 3.2. Performance of ML Algorithms to Predict Health-Related Quality of Life

According to model performance on the test data, the predictive accuracy of the ML models for HRQOL was 0.77 [0.76–0.78] for LR, 0.73 [0.72–0.78] for XGBoost model, and 0.74 [0.73–0.75] for RF model. When predicting physical health, the accuracy was 0.71 [0.70–0.72] for LR, 0.70 [0.68–0.72] for XGBoost, and 0.70 [0.68–0.71] for RF model. For predicting mental health, the accuracy was 0.77 [0.76–0.78] for LR, 0.74 [0.73–0.75] for XGBoost, and 0.72 [0.71–0.73] for RF model (Table 1). The area under the AUC-ROC values ranged between 0.81 and 0.84 for HRQOL, 0.74 and 0.76 for physical health, and 0.83 and 0.86 for mental health (Figure 1, Figure 2 and Figure 3).

### 3.3. Feature Importance

Overall, 87 features were evaluated based on their relevance in predicting HRQOL using the mean decrease accuracy score. Only 20 most important features that influence the prediction accuracy or performance of the ML models were reported. The top five most important features for predicting HRQOL were the feeling that things were going as expected, the ability to stay on top of things, the ability to control irritations, the perception of having no one to rely on for help, feeling God’s presence, and feelings of unhappiness.

In predicting physical health, the most important features were the feeling that things were going as expected, the ability to stay on top of things, the presence of diabetes, the ability to control irritations, and being treated with less courtesy at doctor’s office. For predicting mental health, the top five predictors were feeling nervous or stressed, feeling unhappy, the ability to stay on top of things, the ability to control irritations, and having confidence in handling personal problems (Figure 4, Figure 5 and Figure 6).

## 4. Discussion

Non-medical factors, particularly SDOH, play an important role in influencing the HRQOL of individuals. In this study, we utilized SDOH features to predict HRQOL using three ML models: –LR, XG Boost, and RF. The three models demonstrated strong performance, with accuracies ranging between 0.73 and 0.77 and AUC-ROC values between 0.81 and 0.84 for predicting overall HRQOL. Additionally, these models achieved accuracies between 0.70 and 0.71 and 0.72 and 0.77 for predicting physical and mental health, with AUC-ROC values (AUC-ROC 0.74 to 0.76) for physical health and 0.72–0.77 for mental health (AUC-ROC: 0.83–0.86). The SDOH features were ranked based on their significance/importance to predict HRQOL, and they were comparable across the components of quality of life measures. For instance, the main features that influence HRQOL were the feelings of individuals that things were going as expected, the feeling that people are on top of things, the presence of comorbidities such as diabetes and CKD, feeling God’s presence, feeling that there is no one to help with/rely on, and being able to control irritations in life.

Various studies have also applied ML models to predict HRQOL in different populations. For example, Karri R et al., (2023) employed ML models to predict HRQOL in patients with brain tumors, reporting an AUC-ROC of 0.8 [17], while a Korean study found AUC-ROC of (0.82) for HRQOL, (0.77) for physical health and (0.79) for mental health [14]. Similarly, Liao WW et al., (2022) demonstrated a high predictive performance with an AUC-ROC value of 0.86 [15] in chronic stroke patients. Unlike our study, the previous studies utilized disease specific markers to predict HRQOL that resulted in a slightly higher model performance. Nonetheless, these studies were often limited by small sample sizes and the incorporation of only narrow range of clinical parameters. In our study, the XGBoost model seemed to be more sensitive (0.78) compared to the other models in terms of correctly identifying positive cases. A sensitivity of 78% in our model indicates that the model correctly identifies 78% of the actual positive cases, while potentially missing 23% of them (false negatives), whereas the accuracy of the ML models ranged between 0.73 and 0.77 with the LR model exhibiting higher accuracy. This accuracy level indicates that the model makes correct predictions for 73% to 77% out of every 100 data points. These levels of sensitivity and accuracy might be acceptable in most machine learning applications, though it requires further performance improvement in the healthcare context to correctly identify people who have improved HRQOL. In our study, LR outperformed both RF and XGBoost, which looks an unusual outcome given the general performance advantages of ensemble methods. This may be due to suboptimal hyperparameter tuning for RF and XGBoost, as the optimization process was not exhaustive. Future studies should focus on more systematic and comprehensive hyperparameter tuning to better leverage the capabilities of these models.

Our study identified several key predictors of HRQOL, among which the belief that things are going as expected appeared to be the most frequent predictor of HRQOL. Individuals’ belief that things are going as expected indicates the tendency in which routine activities are going as preplanned. In order to entertain this feeling, a commitment to manage routine/daily activities and to forecast future life events is required. Exposure or engagement in unexpected accidents, crimes, or gambling could cause a decrease in HRQOL. In addition, a disorganized activity and delayed response to urgent issues could lead to mental stress. For instance, a study from a representative sample from the National Health and Nutrition Examination Survey (NHANES) reported that activities of daily living including bathing or showering, dressing, getting in and out of bed or a chair, walking, using the toilet, and eating were associated with HRQOL [12].

The capacity to be on top of things has been reported as the second important predictor of HRQOL. It refers to keeping up with responsibilities, being in control of a situation, and being aware of changes/updates. It may also reflect success or productivity at the workplace or in personal activities. Being on top of things is also linked with optimism. There is evidence that optimistic people who believe that they are capable of executing things present a higher quality of life compared to those with low levels of optimism on things (8). In addition, a recent epidemiologic studies have identified psychosocial assets such as optimism as potential predictors/promotors of longer life [9]. Being on top of things could also suggest success stories of relationships, achievements in education, or earning a good salary. Kivits J et al., (2013) demonstrated that four social indicators, namely living in couple, level of education, occupational status, and net income per household, were determinants of HRQOL [10]. Therefore, it is imperative that the ability to be on top of things can influence HRQOL as a component of SDOH.

Furthermore, emotional control was an important predictor of HRQOL, particularly mental health status. A systematic review of longitudinal observational studies reported that quality of life declined before and during the onset of emotional disorders, such as anxiety and depression, which implied a link between emotional instability/disturbance and quality of life [11]. Emotional dysregulation can exist when there is a complete inability to regulate responses and is a characteristic of several mental health problems, including anxiety, substance abuse, eating disorders, and depression. Consequently, emotionality to internal and external stressors will likely have a direct negative impact on quality of life [32]. Patients with severe emotional disturbance were associated with a lower quality of life as it reduces the accomplishment of their daily tasks, resulting in low self-confidence and self-esteem [33]. A population-based cross-sectional study revealed that among individuals with an emotional problem, the rate of quality of life was lower compared to patients without a history of emotional issues [34]. Also, Silva A et al. (2024) reported a strong association between emotional symptomology and quality of life [35]. The relationship with others, dissatisfaction with income and educational attainment, and low income status might exacerbate emotional distress. Kivits J et al. (2031) demonstrated that four social indicators, namely living in a couple, level of education, occupational status, and net income per household, were determinants of HRQOL, probably through emotional distress [36].

There are different coping strategies to avoid emotional distress/irritability and improve quality of life. For instance, individuals can cope up with painful emotional thoughts and difficult emotions through mindfulness. Mindfulness is an approach which involves the practice of deep focus and concentration on an immediate situation with curiosity and acceptance. This increases the awareness and tolerance of and reduces the responses to emotional experiences, which eventually could lead to improved mental health status [36]. Furthermore, cognitive behavioral therapy (CBT) has also demonstrated effectiveness in patients with excessive emotional dysregulation [37]. CBT is a form of psychological treatment that has been demonstrated to be effective for a range of problems including depression, anxiety disorders, alcohol and drug use problems, marital problems, eating disorders, and severe mental illness. Numerous research studies suggest that CBT leads to significant improvements in functioning and quality of life [38]. In many studies, CBT has been demonstrated to be as effective as, or more effective than, other forms of psychological therapy or psychiatric medications, allowing people to learn to recognize one’s distortions in thinking that are creating problems and then to re-evaluate them while considering reality. CBT treatment usually involves gaining a better understanding of the behavior and motivation of others, using problem-solving skills to cope with difficult situations, and learning to develop a greater sense of confidence in one’s own abilities [39]. These behavioral activities could eventually lead to a change in the HRQOL of individuals.

Additionally, our study revealed that feeling God’s presence has been reported as an important variable that might influence HRQOL. A belief in God’s presence refers to reliance in divine intervention to accomplish things. People tend to overcome challenges in their daily life through divine support. A related systematic review has indicated that religiously/spiritually higher levels among adults were associated with higher HRQOL levels and has suggested that religiosity/spirituality can be an important strategy to cope with adverse situations, irrespective of the medical condition of individuals [40]. It is contemplated that people with high spirituality and religiosity have a positive correlation with their environment, psychological, social relationships, and overall quality of life domains [41]. Religious people seem to worry less when they encounter a problem by relying on God and giving life a different meaning. This self-defense mechanism might ultimately avoid worries and stress, accompany by improved HRQOL [42].

In general, our study demonstrated the crucial role of SDOH in predicting HRQOL using ML techniques. We incorporated a comprehensive set of more than SDOH features with different rating scales, overcoming the limitations of previous studies that only employed a limited number of socioeconomic variables and clinical features in predicting HRQOL. The inclusion of a large number of subjects (~ 100,000) in the testing and training of the model lends credibility to our model’s performance. Our study presents a novel approach to predict HRQOL, which can be applied to all individuals reporting SDOH, regardless of their underlying medical conditions. Moreover, the inclusion of more than 80% of participants who have been historically underrepresented in biomedical research (i.e., individuals with inadequate access to medical care, >65 years, annual income below 200% of the federal poverty level (FPL), disability, less than a high school education or equivalent) makes our findings generalizable/translatable to wide range of underserved populations [43].

Nonetheless, while interpreting the findings, the following limitations of our study should be considered. Firstly, patients were requested to rate their HRQOL, including physical and mental health status, on a Likert scale, which was later converted to dichotomous variable. In future studies, a standard HRQOL assessment tool that has good reliability and validity could be incorporated into the All of Us survey [44]. Furthermore, our study tested a limited number of ML algorithms. Additional models could be trained in the future studies, which might result in improved prediction performance, because there is a good chance of improving the sensitivity, precision, and accuracy of the models considering the current performance scores. Moreover, future studies can implement explainable AI methods to explore the direction of an SDOH feature’s impact.

## 5. Conclusions

In conclusion, this study highlights the significant influence of social determinants of health (SDOH) on HRQOL, encompassing social, educational, healthcare, emotional, and environmental factors. By using SDOH data, our ML models demonstrated enhanced prediction performance for HRQOL, underscoring the importance of these non-medical factors in determining health outcomes. The findings suggest that SDOH can effectively predict HRQOL across diverse population, regardless of existing medical conditions.

To measure HRQOL, strategies such as assessing individuals’ expectations, evaluate daily routines, level of emotional regulation, and optimism can be implemented. Additionally, spirituality and religion, where culturally relevant, may provide meaningful contributions to predicting HRQOL. As such, a routine evaluation of SDOH during clinical visits could prove instrumental in identifying patient needs and improving overall well-being.

Future research should focus on integrating more comprehensive HRQOL assessment tools within large datasets like All of Us and exploring additional ML models to further refine predictive accuracy. By doing so, the hierarchical significance of SDOH on HRQOL prediction can be better understood, and interventions designed to target specific SDOH factors could be developed to enhance HRQOL across a broader range of populations.

## Figures and Tables

**Figure 1 bioengineering-12-00166-f001:**
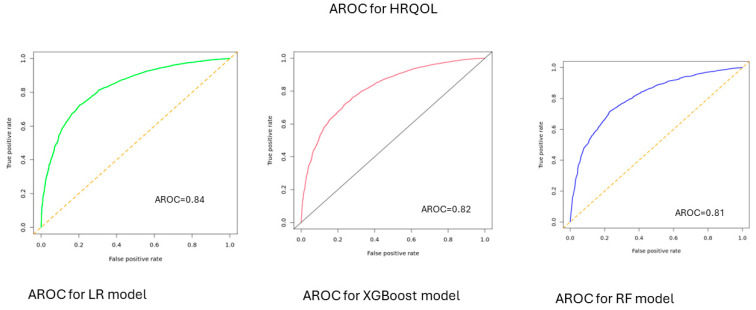
AUC-ROC for HRQOL, All of Us participants, 2021–2022.

**Figure 2 bioengineering-12-00166-f002:**
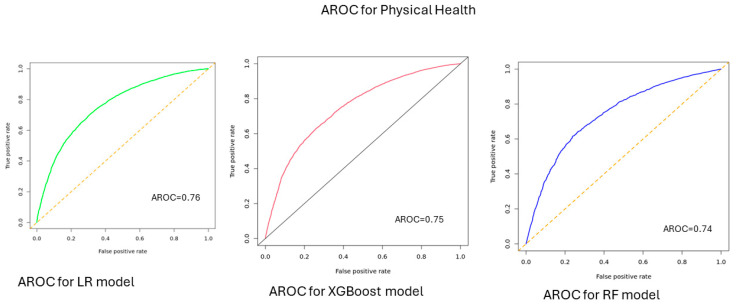
AUC-ROC for physical health, All of Us participants, 2021–2022.

**Figure 3 bioengineering-12-00166-f003:**
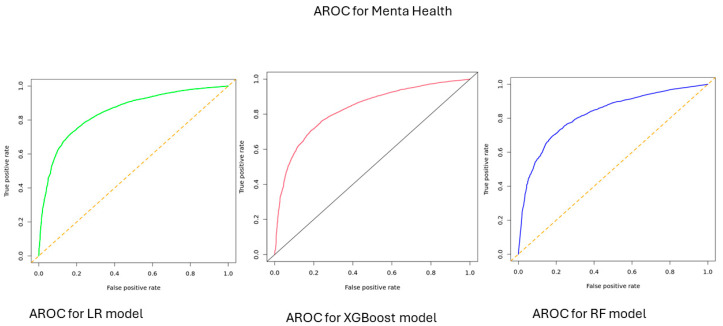
AUC-ROC for mental health, All of Us participants, 2021–2022.

**Figure 4 bioengineering-12-00166-f004:**
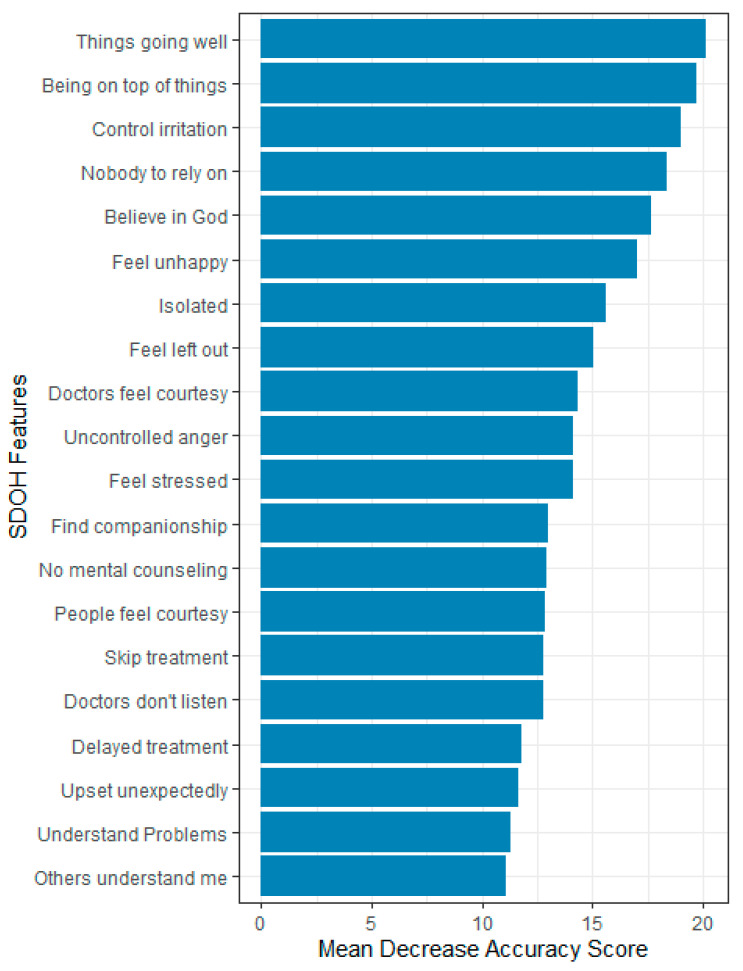
Feature importance for HRQOL, All of Us participants, 2021–2022.

**Figure 5 bioengineering-12-00166-f005:**
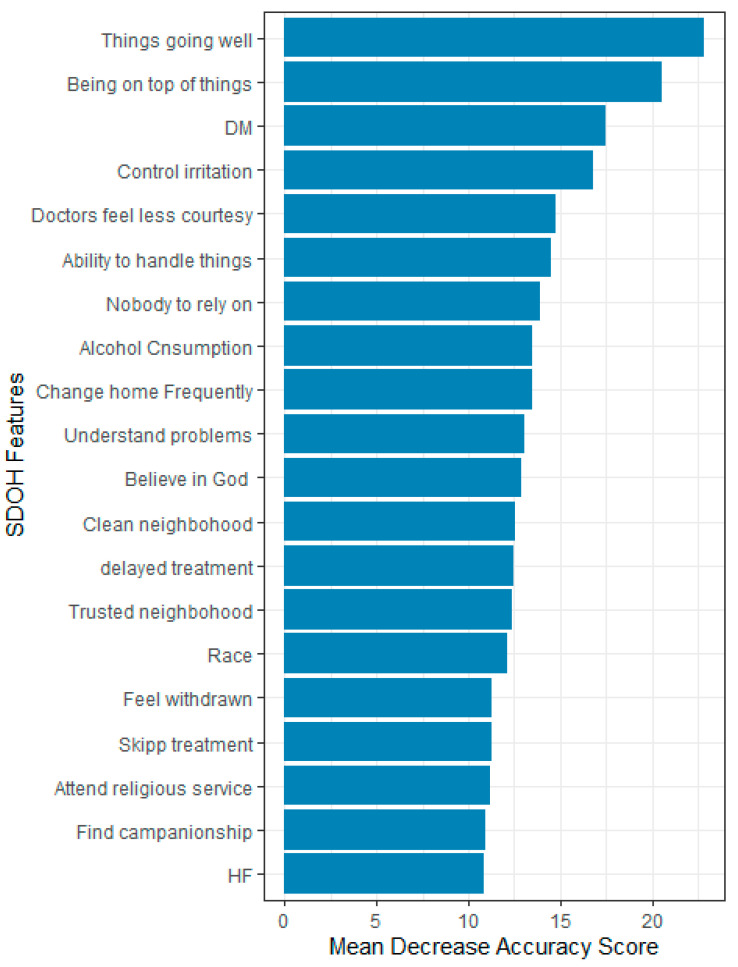
Feature importance for physical health, All of Us participants, 2021–2022.

**Figure 6 bioengineering-12-00166-f006:**
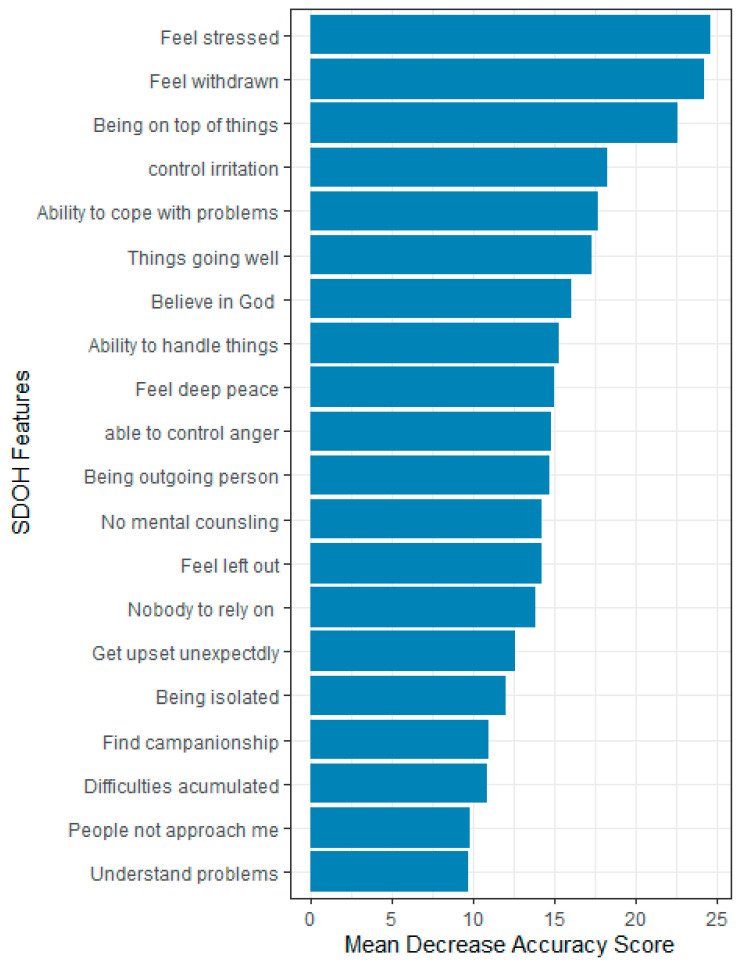
Feature importance for mental health, All of Us participants, 2021–2022.

**Table 1 bioengineering-12-00166-t001:** Performance of ML algorithms to predict health-related quality of life in All of Us Cohort.

HRQOL					
Models	AUC-ROC	Sensitivity	Precision	F-1	Accuracy
LR	0.84	0.75	0.77	0.76	77 [0.76–0.78]
XGBoost	0.82	0.78	0.73	0.75	0.73 [0.72–0.74]
RF	0.81	0.77	0.74	0.75	0.74 [0.73–0.75]
**Physical Health**					
Models	AUC-ROC	Sensitivity	Specificity	F-1	Accuracy
LR	0.76	0.66	0.72	0.70	0.71 [0.70–0.72]
XGBoost	0.75	0.76	0.61	0.67	0.70 [0.68–0.72]
RF	0.74	0.79	0.60	0.68	0.70 [0.68–0.71]
**Mental Health**					
Models	AUC-ROC	Sensitivity	Specificity	F-1	Accuracy
LR	0.85	0.78	0.76	0.77	0.77 [0.76–0.78]
XGBoost	0.83	0.80	0.72	0.76	0.74 [0.73–0.75]
RF	0.83	0.80	0.72	0.76	0.72 [0.71–0.73]

## Data Availability

The original contributions presented in this study are included in the article/Appendix A. Further inquiries can be directed to the corresponding author(s).

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
