# Peer review of "Predicting Health-Related Quality of Life Using Social Determinants of Health: A Machine Learning Approach with the All of Us Cohort"

_bioengineering, 2025, doi:10.3390/bioengineering12020166_

Round 1

Reviewer 1 Report (Previous Reviewer 1)

Comments and Suggestions for Authors

Dear respected authors,

thank you for considering my comments. However, the justificiation for the choice of algorithms does not convince me yet. You are saying " We preferred to employ tree-based models because they offer a clear visual representation of decision-making, handle missing data well, and can be easily explained. In addition, These ML algorithms have a high prediction accuracy...", but you are not making use of the explainability, neither of the visual representation. Moreover, you later say that the results should be improved for clinical use. You feature importance analysis can be done with arbitrary type of algorithms. Please clearly justify why you used the algorithms and then make use of their specific features and exploit their benefits.

Author Response

Reviewer 1

Reviewer comment: Thank you for considering my comments. However, the justificiation for the choice of algorithms does not convince me yet. You are saying " We preferred to employ tree-based models because they offer a clear visual representation of decision-making, handle missing data well, and can be easily explained. In addition, These ML algorithms have a high prediction accuracy...", but you are not making use of the explainability, neither of the visual representation. Moreover, you later say that the results should be improved for clinical use. You feature importance analysis can be done with arbitrary type of algorithms. Please clearly justify why you used the algorithms and then make use of their specific features and exploit their benefits.

Author response

Dear Reviewer,

Thank you for your insightful comments and for highlighting the need for a more robust justification of our algorithm selection. We value your feedback, and we have carefully addressed your concerns as outlined below:

We appreciate your point about the need for a clearer rationale for choosing tree-based models. While our initial justification mentioned their explainability, handling of missing data, and high prediction accuracy, we recognize that we did not fully demonstrate or leverage these features in the manuscript. To address this, we have revised the justification as follows:

“Tree-based models were selected for their ability to inherently model non-linear relationships and capture complex interactions between variables, which are common in clinical datasets. They provide a robust performance baseline due to their flexibility and capability to handle both numerical and categorical data effectively. Additionally, ensemble tree-based methods have demonstrated strong performance in clinical predictive modeling tasks [166-170]”.

We have also elaborated on these points in the revised manuscript and included relevant references to support our choice [ref 29].

We acknowledge your observation that the results need to be improved for clinical use. In the revised manuscript, we have addressed how the insights from feature importance analysis can help to enhance predictive performance in clinical practice [lines 396-402].

We hope the revised manuscript provides a stronger justification for our algorithm choice and better demonstrates how their specific features are utilized.

Reviewer 2 Report (Previous Reviewer 3)

Comments and Suggestions for Authors

Abstract

It is not the way we write abstracts!

I have never seen "Abstract Background:" in any abstract and I have seen them a lot in my 40+ years of publishing activity.

It is full of numbers and abreviations.It creates clutter and DISCOURAGES people from possibly reading and citing your submission.

Is it what you hope to achieve? 

---

Please check:  harmonic mean (1)

with Wikipedia. It looks a bit strange.

It should be the reciprocal of the arithmetic mean of the reciprocals. I am not claiming that it is wrong, but it seems somehow strange, and the formula should be expanded to the definition in Wikipedia, followed by your "simplified" version. Explanations for using this strange meaning should be provided.

AROC is not used but AUC of ROC.

Author Response

Reviewer 2

 Reviewer Comment 1: It is not the way we write abstracts! I have never seen "Abstract Background:" in any abstract and I have seen them a lot in my 40+ years of publishing activity.

Author response

Dear Reviewer,

Thank you for your valuable feedback on the abstract. We appreciate your suggestions and have revised the abstract to align with bioengineering journal requirements. The abstract now follows a single-paragraph format, presenting the study’s background, methods, results, and conclusions in a logical flow without explicit headings.

Reviewer Comment 2: It is full of numbers and abreviations. It creates clutter and DISCOURAGES people from possibly reading and citing your submission. Is it what you hope to achieve?  

Author response

Dear Reviewer,

Thank you for your valuable feedback regarding the numbers and abbreviations of the abstract. We appreciate your concern about excessive numbers and abbreviations potentially discouraging readers. Based on your suggestion, we have revised the abstract to enhance readability and clarity while retaining the critical information necessary to convey the study's findings. We reduced the use of abbreviations and included only commonly recognized terms (e.g., ML, HRQOL, SDOH), which were clearly defined at their first mention.

The revised abstract now reads as follows:

“This study applied machine learning (ML) algorithms to predict health-related quality of life (HRQOL) using comprehensive social determinants of health (SDOH) features. Data from the All of Us dataset, including participants with complete HRQOL and SDOH records, were analyzed. The primary outcome, HRQOL, encompassed physical and mental health components, while SDOH features included social, educational, economic, environmental, and healthcare access factors. Three ML algorithms: logistic regression, XGBoost, and Random Forest, were tested. The models achieved accuracy ranges of 0.73–0.77 for HRQOL, 0.70–0.71 for physical health, and 0.72–0.77 for mental health, with corresponding AUC ranges of 0.81–0.84, 0.74–0.76, and 0.83–0.85, respectively. Emotional stability, activity management, spiritual beliefs, and comorbidities were identified as key predictors. These findings underscore the critical role of SDOH in predicting HRQOL and suggest that future research should focus on applying such models to diverse patient populations and specific clinical condition [lines 15-21]”

Reviewer comment 3: Please check:  harmonic mean (1).

It should be the reciprocal of the arithmetic mean of the reciprocals. I am not claiming that it is wrong, but it seems somehow strange, and the formula should be expanded to the definition in Wikipedia, followed by your "simplified" version. Explanations for using this strange meaning should be provided.

Author response

Dear Reviewer,

Thank you for pointing out the need to clarify the use of the harmonic mean in our manuscript. Specifically, you highlighted the context in which it is applied. The F1 score, as we used in this study, is indeed the harmonic mean of precision and recall. We have revised the manuscript to better explain this relationship and to ensure clarity. Below are the details of our updates:

F1    Precision  

TP: True positives, FP: False positive, FN: False negative

The harmonic mean is used here because it effectively balances precision and recall, penalizing significant differences between the two. This makes it a suitable metric for evaluating classification models, particularly in cases where data is imbalanced [lines 205-209].

Reviewer comment 5: with Wikipedia. It looks a bit strange.

Author response:

Thank you for pointing this out. We acknowledge that Wikipedia is not a primary or scholarly source and may not meet the standards of academic referencing. To address this, we have replaced the Wikipedia citation (reference 4) with [Braveman P, Gottlieb L. The social determinants of health: it's time to consider the causes of the causes. Public Health Rep. 2014 Jan-Feb;129 Suppl 2(Suppl 2):19-31. doi: 10.1177/00333549141291S206. PMID: 24385661; PMCID: PMC3863696] to ensure the reliability and credibility of the referenced information.

Reviewer comment 6: AROC is not used but AUC of ROC.

Thank you for pointing this out. We agree with the reviewer that "AUC-ROC" is the appropriate terminology. We have reviewed the manuscript and replaced any instances of "AROC" with "AUC-ROC" to ensure accuracy and consistency in terminology throughout the text. We appreciate your attention to detail and have made this correction to improve clarity.

Reviewer 3 Report (Previous Reviewer 2)

Comments and Suggestions for Authors

The authors analyse the prediction of the HRQOL improvement on the All of Us dataset from the SDOH survey data. This is a binary classification problem with discrete multiclass inputs. The paper had some issues with the lack of information in the first submission, but all the concern of the revision has been solved in this reviewed version.

The methodology is adecuated for this problem, and the conclusions of the study are supported by the presented results.

Author Response

Reviewer 3

Reviewer Comments

The authors analyse the prediction of the HRQOL improvement on the All of Us dataset from the SDOH survey data. This is a binary classification problem with discrete multiclass inputs. The paper had some issues with the lack of information in the first submission, but all the concern of the revision has been solved in this reviewed version. The methodology is adecuated for this problem, and the conclusions of the study are supported by the presented results.

Author Response:
Thank you for your positive and encouraging feedback. We are glad to hear that the revisions addressed the previous concerns and that you found the methodology appropriate and the conclusions well-supported by the results. We appreciate your constructive review, which has significantly improved the quality of our work.

Round 2

Reviewer 1 Report (Previous Reviewer 1)

Comments and Suggestions for Authors

Dear authors,

thank you for addressing all of my comments. I do not have any more.

This manuscript is a resubmission of an earlier submission. The following is a list of the peer review reports and author responses from that submission.

Round 1

Reviewer 1 Report

Comments and Suggestions for Authors

Dear respected authors,

thank you very much for your interesting work on the AoU dataset and predicting the QoL. I have a few comments on your work.

1. Comment

Please proofread your manuscript a second time. I found some spelling and grammar mistakes. Here are a some examples:

l.86: redundant space

Eq.1: missing i's

2. Comment:

I like the structure of your abstract. However, I recommend to add line breaks before results, conclusion,...

3. Comment

Also regarding your abstract, think about skipping the intervals and refrain from using abbreviations.

4. Comment

I did not understand what is the novelty of your approach. You simply applied common ML algorithms to a dataset. I suggest to explicitly state the novelty of your approach or findings.

5. Comment

In the sense of my previous comment, I am missing a state of the art section. There you can describe recent approaches and their gaps you are addressing.

6. Comment

Can you please provide more details on the study of the AoU dataset, e.g., frequency, time between measurements etc.?

7. Comment

As I mentioned before, you are applying standard ML algorithms. The justification for choosing them is too shallow. Can you please elaborate why you did not choose other methods like neural networks?

8. Comment

In eq. 1 you write F-1, that is not the correct spelling. You could interpret the - as subtract operator. It is F subscript 1.

9. Comment

Did you use stratified splitting for your data?

10. Comment

How did you optimise the hyperparameters? I do not understand your explanation in line 142/143.

11. Comment

It is not clear what your classifiers were trained on. What are the features and what are the labels?

12. Comment

Was your dataset imbalanced? If yes, how did you deal with it?

13. Comment

The AUC metric is defined for binary classification. Was your task multi-class classification and if so, how did you adapt the metric?

14. Comment

Feature importance and explainable AI methods are really important, especially, im the medical context. Why did you use a simple one and not a more elaborate method like Shapley values?

15. Comment

Your discussion section seems a bit unfocused and too shallow to me. At first, you are just describing your results and later you are not discussing weaknesses of the ML methods and their impact. You just state they have strong performance, which is questionable as well. I suggest to revise the discussion section and discuss the algorithms as well.

16. Comment

Regarding the discussion, you are introducing a lot new references and aspects there. I think that would be a good fit for a state of the art section. I suggest to introduce the research in the SOTA or introduction section.

17. Comment

The data availability statement at the end is from the template. Please use the statement that fits to your data.

18. Comment

I feel your acknowledgements are too broad. I suggest to explicitly name individuals, but I will leave that to the editor's discretion.

19. Comment

I recommend to release your source code, at least to the reviewers. That would enable them to validate your results.

Reviewer 2 Report

Comments and Suggestions for Authors

This paper presents an ML classifier on health-related quality of life (HRQOL) using comprehensive social determinants of health (SDOH) features from the All of Us (AoU) dataset. The authors use three well-known ML methods (LR, RF, and XGBoost) without any novelty or improvement over their standard implementations.

The paper is well presented, and the methodology has no flaws. However, the work has very limited usefulness and does not contribute any scientific improvement.

Moreover, I have some doubts about the work that need clarification:

- The AoU dataset has four health-related outcomes: general health status, HRQOL, physical health, and mental health. Why does the paper use only the first three?

- It is unclear which input variables are used in the models. Are all variables discrete? Are they dichotomous or multiclass? Are there any continuous variables? The authors should include a description of the input variables.

- Are the performances in Table 1 measured on the test split? Which output category is used as the positive class for the metrics: the improved or unimproved class?

- Regarding feature importance (Section 3.3), each model should have its own importance. Are the presented values the average of the three ML models, or from only one of them?

- It seems very strange that the LR method shows the best performance of the three models, as RF and XGBoost usually outperform LR by a large margin. This could be because of poor hyperparameter selection, which is not clearly described in the paper. At least, the authors should include a brief comment on this behavior in the Discussion.

Reviewer 3 Report

Comments and Suggestions for Authors

> original text (for Ctrl-F)

R: reviewer

> "social determinants of health"

R: used in abstract and references only.

Judging by the number of Ctrl-F hits, it is a fundamental concept! It needs to be used (not as acronym) in Intro and Conclusions (with the acronym followed it).

R: Reference to:

https://en.wikipedia.org/wiki/Social_determinants_of_health

is a must!

R: the same goes for "All of Us Cohort"

R: is it: https://allofus.nih.gov/get-involved/participation

R: rating scale is used once only.

R: ading https://www.who.int/health-topics/social-determinants-of-health#tab=tab_1 is highly advisable

R: the submission uses plenty of acronyms. Appedix is needed to list their meaning.

Conlusions

> In summary,

R: remove (sereves no purpose)

> By integrating SDOH data

R: I could not see "integration" and if it is, it should be:

R By integrating SDOH data with...

Future research should include: 

since the quality of life is strongly associated with using rating scales and it is not well-specified in the submission. De facto, every supervised ML dataset is a rating scale.